# TOWARDS THE VULNERABILITY OF WATERMARKING ARTIFICIAL INTELLIGENCE GENERATED CONTENT

## ABSTRACT

Artificial Intelligence Generated Content (AIGC) is gaining great popularity in social media, with many commercial services available. These services leverage advanced generative models, such as latent diffusion models and large language models, to generate creative content (e.g., realistic images, fluent sentences) for users. The usage of such generated content needs to be highly regulated, as the service providers need to ensure the users do not violate the usage policies (e.g., abuse for commercialization, generating and distributing unsafe content). A promising solution to achieve this goal is watermarking, which adds unique and imperceptible watermarks on the content for service verification and attribution.

Numerous watermarking approaches have been proposed recently. However, in this paper, we show that an adversary can easily break these watermarking mechanisms. Specifically, we consider two possible attacks. (1) Watermark removal: the adversary can easily erase the embedded watermark from the generated content and then use it freely without the regulation of the service provider. (2) Watermark forge: the adversary can create illegal content with forged watermarks from another user, causing the service provider to make wrong attributions. We propose WMaGi, a unified framework to achieve both attacks in a holistic way. The key idea is to leverage a pre-trained diffusion model for content processing, and a generative adversarial network for watermark removing or forging. We evaluate WMaGi on different datasets and embedding setups. The results prove that it can achieve high success rates while maintaining the quality of the generated content. Compared with existing diffusion model-based attacks, WMaGi is **5,050∼11,000×** faster.

## 1 INTRODUCTION

Benefiting from the advance of generative deep learning models (Rombach et al., 2022; Touvron et al., 2023), Artificial Intelligence Generated Content (AIGC) has become increasingly famous. Many commercial services have been released, which leverage large models (e.g., ChatGPT (cha), Midjourney (Mid)) to generate creative content based on users' demands. The rise of AIGC also leads to some legal considerations, and the service provider needs to set up some policies to regulate the usage of generated content. *First*, the generated content is one important intellectual property of the service provider, many services do not allow users to make the AIGC into commercial use (Touvron et al., 2023; Mid). Selling the generated content for financial profit [1] will violate this policy and cause legal issues. *Second*, generative models have the potential of outputting unsafe content (Wei et al., 2023; Qi et al., 2023; Liu et al., 2023a; Le et al., 2023), such as fake news (Guo et al., 2021), malicious AI-powered images (Salman et al., 2023; Le et al., 2023), phishing campaigns (Hazell, 2023), and cyberattack payloads (Charan et al., 2023). New laws are established to regulate the generation and distribution of content from deep learning models on the Internet[234].

As protecting and regulating AIGC become urgent, Google hosted a workshop in June 2023 to discuss the possible solutions against malicious usage of generative models (Barrett et al., 2023). Not surprisingly, the *watermarking* technology as mentioned as a promising defense. By adding

---

[1]https://okuha.com/best-sites-to-sell-ai-art/

[2]https://www.reuters.com/technology/governments-efforts-regulate-ai-tools-2023-04-12/

[3]https://www.pdpc.gov.sg/help-and-resources/2020/01/model-ai-governance-framework

[4]https://www.lexology.com/library/detail.aspx?g=42ad7be8-76bd-40c8-ae5d-271aaf3710eb

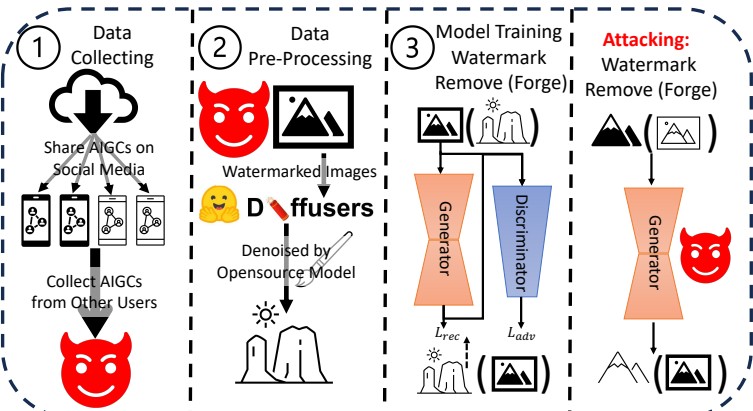

Figure 1: Overview of `WMaGi`. (1) The adversary collects data from the target AIGC service. (2) The adversary uses an open-source denoising model to purify the collected data. (3) The adversary adopts the original and purified data to train a GAN, which can be used to remove or forge the watermark. Black and white images stand for images with and without watermarks, respectively.

invisible specific watermarks to the generated content (Fernandez et al., 2023; Kirchenbauer et al., 2023; Liu et al., 2023b), we are able to identify the misuse of AIGC and track to the corresponding users. A variety of robust watermarking methodologies have been designed, which can be classified into two categories. (1) A general method is to make the generative model learn a specific data distribution, which can be decoded by another deep learning model to obtain a secret message as the watermark (Fernandez et al., 2023; Liu et al., 2023b; Zhao et al., 2023b). (2) The model owner can concatenate a watermark embedding model (Zhu et al., 2018; Tancik et al., 2020) after the generative model to make the final output contain watermarks. A very recent work from DeepMind, SynthID Beta (Syn), detects AI-generated images by adding watermarks to generated images[5]. According to its description, this service possibly follows a similar strategy as StegaStamp (Tancik et al., 2020), which adopts an encoder to embed watermarks into images and a decoder to identify the embedded watermarks in the given images.

The Google workshop (Barrett et al., 2023) reached the consensus that "existing watermarking algorithms only withstand attacks when the adversary has no access to the detection algorithm", and embedding a watermark to a clean image or text "seems harder for the attacker, especially if the watermarking process involves a secret key". However, in this paper, we argue that it is not the case. We find that it is easy for an adversary without any prior knowledge to **remove** or **forge** the embedded secret watermark in AIGC, which will break the IP protection and content regulation. Specifically, (1) a watermark removal attack makes the service providers fail to detect the watermarks it embeds into the AIGC previously, so the malicious user can circumvent the policy regulation and abuse the content for any purpose. (2) A successful watermark forge attack can intentionally embed a watermark of another user into the unsafe content without the knowledge of the secret key. This could lead to wrong attributions and frame up that benign user.

We introduce `WMaGi`, a novel framework to achieve both watermark removal and forge attacks against AIGC in a unified manner. The key idea is to leverage a pre-trained diffusion model and train a generative adversarial network (GAN) for erasing or embedding watermarks to AIGC. Figure 1 shows the overview of `WMaGi`, which consists of three steps: data collection, data pre-processing, and model training. In the first step, the adversary collects AIGC from the target service or a specific user. We assume that the adversary can only collect the watermarked AIGC, without any clean content. Furthermore, we assume that the same secret watermark message is applied to all the collected data. More details of our threat model can be found in Section 3.2. In the second step, we introduce an adversarial pipeline to weaken the embedded watermark messages in the data. Specifically, the adversary adopts a public diffusion model, such as DDPM (Ho et al., 2020), to denoise the collected data. The diffusion model can be either non-watermarked, or watermark-protected with a different secret. Its preprocessing operation can make the embedded message unrecoverable from the denoised data. In the third step, the adversary trains a GAN model to map the data distribution from collected data to denoised data (for watermark removal) or from denoised data to collected data (for watermark

---

[5]Up to the date of writing, SynthID Beta is still a beta product only provided to a small group of users. Unfortunately, we do not have access to it. Therefore, we cannot provide evaluation results with respect to it in our experiments.

forging). After the model is trained, the adversary can adopt the generator to remove or forge the specific watermark for AIGC, depending on the target in the third step.

We evaluate our proposed `WMaGi` on various datasets (e.g., CIFAR-10, CelebA), and settings (e.g., different watermark lengths, few-shot learning), to show its generalizability. Our results prove that the adversary can successfully remove or forge a specific watermark in the AIGC and keep the content indistinguishable from the original one. This provides concrete evidence that existing watermarking schemes are not reliable, and the community needs to explore more robust watermarking methods. Overall, our contribution can be summarized as follows:

- To the best of our knowledge, it is the first work focusing on removing and forging watermarks in AIGC under a black-box threat model. Furthermore, `WMaGi` is a unified framework, which can achieve both attack goals in a holistic way. Our study discloses the unreliability and fragility of existing watermarking schemes.

- Different from prior attacks, `WMaGi` does not require the adversary to have clean data or any information about the watermarking schemes, which is more practical in real-world applications.

- Comprehensive evaluation proves that `WMaGi` can remove or forge the watermarking information without harming the data quality. `WMaGi` is time-efficient, which is 5,050∼11,000× faster than existing attacks with diffusion models.

- We prove that `WMaGi` is effective in the few-shot setting, i.e., it can be freely adapted to unseen watermarks. Furthermore, `WMaGi` remains highly effective for different watermark lengths.

## 2 RELATED WORKS

### 2.1 CONTENT WATERMARK

Driven by the rapid development of large and multi-modal models, there is a renewed interest in generative models, such as ChatGPT (cha) and Stable Diffusion (Rombach et al., 2022), due to their capability of creating high-quality images (Ho et al., 2020; Rombach et al., 2022), texts (cha; Touvron et al., 2023), audios (Kong et al., 2021), and videos (Ho et al., 2022). The generated content is referred to Artificial Intelligence Generated Content (AIGC). Such AIGC can have high IP values and sensitive content. Therefore, it is important to protect and regulate it during its distribution on public platforms, e.g., Twitter (Twi) and Instagram (Ins).

A typical strategy to achieve the above goal is watermarking: the service provider adds a secret and unique message to the content, which can be subsequently extracted for ownership verification and attribution. Existing watermarking schemes can be divided into post hoc methods and prior methods. Post hoc methods convert the clean content into watermarked content following one of the following two strategies. (i) Visible watermark strategy: the service provider adds characters or paintings into the clean content (Liu et al., 2021; Cheng et al., 2018), which can be recognized by humans. (ii) Invisible watermark strategy: the service provider embeds a specific bit string into the clean content by a pre-trained steganography model (Zhu et al., 2018; Tancik et al., 2020) or signal transformation (Nam et al., 2021), which will be decoded by a verification algorithm later.

For prior methods, the generative model directly learns a distribution of watermarked content, which can be decoded by a verification algorithm (Fei et al., 2022; Fernandez et al., 2023; Cui et al., 2023; Zhao et al., 2023b). Specifically, Fei et al. (2022) designed a watermarking scheme for generative adversarial networks (GANs), by learning the distribution of watermarked images supervised by the watermark decoder. Fernandez et al. (2023); Zhao et al. (2023b) designed a watermarking scheme for diffusion models (Rombach et al., 2022), which embeds a predefined bit string into the generated images. The bit string can be restored with a secret decoder. Therefore, the service provider can recognize the AIGC from his generative model or determine the specific user account.

In this paper, we target both post hoc methods and prior methods. For post hoc methods, we do not consider visible watermarks as they can significantly decrease the visual quality of AIGC, making them less popular for practical adoption. For invisible watermarks, we only consider the steganography approach, as it is much more robust and harder to attack than the signal transformation approach (Nam et al., 2021; Wang et al., 2022; Zhao et al., 2023a).

## 2.2 ATTACKS AGAINST WATERMARKS

To the best of our knowledge, there is no work considering the watermark forge attack. Prior efforts mainly focus on the watermark removal attack. These attack solutions can be summarized into three main categories, i.e., image inpainting methods (Ulyanov et al., 2018; Liang et al., 2021) for visible watermarks, denoising methods (Li, 2023; Zhao et al., 2023a), and disrupting methods (Nam et al., 2021; Wang et al., 2022) for invisible watermarks. However, they have several critical drawbacks in practice. Specifically, the image inpainting methods (Ulyanov et al., 2018; Liang et al., 2021) require clean images and watermarked images to train the inpainting model, which is not feasible in the real world, because the user can only obtain watermarked images from the service providers (Mid). Disrupting methods (Nam et al., 2021; Wang et al., 2022) require the user to know the details of the watermarking schemes, which is also difficult to achieve. The most promising method is based on denoising models. For instance, Li (2023) adopted guided diffusion models to purify the watermarked images and minimize the differences between the watermarked images and diffusion model's outputs. However, using diffusion models to remove the watermark will cost a lot of time.

In sum, compared to prior works, (1) **we are the first to consider the watermark forge attack**; (2) **we build a unified framework `WMaGi` to realize these two types of attacks in a holistic way**; (3) **our watermark removal attack from `WMaGi` is more practical as it does not require the clean content or watermarking schemes. It is also more efficient as it brings 5,050$\sim$11,000$\times$ speedup compared to diffusion model-based attacks**.

## 3 WMaGi: A UNIFIED ATTACK FRAMEWORK

In this section, we first give a formal definition of the watermark verification process. Then, we introduce our threat model in the context of adversary's power and knowledge. Finally, we introduce details of our proposed `WMaGi`. To the best of our knowledge, it is the first black-box watermark removal and forging method in practice. We mainly consider watermarks embedded in the generated images. Watermarks in other other domains, such as language and audio, will be our future work.

### 3.1 PRELIMINARY

We consider a general watermarking scheme, which is widely studied in previous works (Fernandez et al., 2023; Liu et al., 2023b) and can be used to protect and regulate the AIGC. In this scheme, the service provider adopts a decoder $\mathcal{M}_D$ to recover the embedded secret message $m$, i.e., the watermark, from the image $x$ generated by its model $\mathcal{M}_G$. For a successful watermark verification, we give the following definition:

**Definition 1** *Given a threshold $\tau$, a message decoder $\mathcal{M}_D$, and a secret message $m$, if an image $x^s$ fulfills $\mathrm{Dis}(\mathcal{M}_D(x^s), m) \leq f(\tau, m)$, where $\mathrm{Dis}(\cdot, \cdot)$ is a distance metric and $f(\cdot, \cdot)$ is a function of $\tau$ and $m$, we say that $x^s$ passes the verification with respect to the secret message $m$. Otherwise, verification fails.*

Definition 1 is general for all types of secret message $m$. Specifically, the most popular type used in many watermarking implementations is a bit string (Fei et al., 2022; Fernandez et al., 2023; Cui et al., 2023; Zhao et al., 2023b). When $m$ is a bit string, the distance $\mathrm{Dis}(\cdot, \cdot)$ is the Hamming Distance, and $f(\tau, m) = \tau \times |m|$, where $|m|$ represents the length of the message $m$. In this paper, we mainly consider this type of watermark message, which is the most reliable and mature method.

### 3.2 THREAT MODEL

**Attack Goals.** We consider a practical scenario where a service provider employs a large generative model $\mathcal{M}_G$ to generate creative images for public users. The service provider embeds a secret user-specific watermark $m$ to each generated image. By extracting the watermark $m$ from any misused image on the Internet, the service provider is able to detect the policy violation events and attribute to the corresponding users.

A malicious user can break this watermarking scheme with two distinct goals. (1) **Watermark removal attack**: the adversary receives a generated image from the service provider, which contains the secret watermark associated with him. He aims to erase the watermark from the generated image,

and then use it freely without the consideration of following the service policy, as the provider is not able to identify the watermarks and track to him any more. (2) **Watermark forge attack**. The adversary tries to frame up a victim user by forging the victim's watermark on a malicious image (from another model or created by humans). Then the adversary can distribute the image on the Internet. The service provider will attribute to the wrong user.

**Adversary's capability.** We consider the black-box scenario, where the adversary can only obtain the generated image $x$ and has no knowledge of the employed generated model or watermark scheme. This is practical, as many service providers only release APIs for users to use their models without leaking any information about the details of the backend models $\mathcal{M}_G$ and $\mathcal{M}_D$. We further assume that all the generated images from the target service are watermark-protected, so the adversary cannot collect any clean images. These assumptions increase the attack difficulty compared to prior works.

### 3.3 METHODOLOGY DETAIL

We introduce `WMaGi` to manipulate watermarks with the above goals. To overcome the black-box challenges, the adversary can adopt a pre-trained denoise model, which accepts noisy images as its inputs and returns new images containing less noise. Then he can train a GAN model to remove or forge watermarks. The details of our `WMaGi`, comprised of three steps, are described as follows.

**Step 1: Data Collection.** The adversary collects the images $x_i$ generated by the target service provider for the target user from the Internet. Such data collection is feasible, as people who share their created content on social media are normally using a specific account and adding tags to indicate the used service. Alternatively, the adversary can also query the service to collect the watermarked images with his account. All the collected data contain one specific watermark $m$. This establishes a dataset $\mathcal{X} = \{x_i | x_i \sim (\mathcal{M}_G, m)\}$, where $\mathcal{M}_G$ is the service provider's generative model.

**Step 2: Data Pre-processing.** Then, the adversary needs to modify the collected data to weaken the embedded watermarks. He can adopt a pre-trained denoise model $\mathcal{H}$, which can be downloaded from a public source, like Hugging Face. The adversary creates a new dataset $\hat{\mathcal{X}} = \{\hat{x}_i | \mathcal{H}(x_i) = \hat{x}_i, x_i \in \mathcal{X}\}$, and uses $\hat{x}_i$ to approximate the clean image $\tilde{x}_i$. Note that it is possible that $\mathcal{H}(x)$ will be visually different from $x$. Our method does not implicitly constrain the similarity between $\mathcal{H}(x)$ and $x$, making it more general.

**Step 3: Model Training.** The adversary needs to build a connection between clean images and watermarked images, so that he can remove or forge a specific watermark. To find such a map, he can adopt a generative adversarial network (GAN) with $\mathcal{X}$ and $\hat{\mathcal{X}}$. There are two components in training the GAN model, i.e., generator $\mathcal{G}$ and discriminator $\mathcal{D}$. Specifically, for watermark removal, $\mathcal{G}$ is to modify $x_i$, and $\mathcal{D}$ is to judge the distribution similarity between $\mathcal{G}(x_i)$ and $\hat{x}_i$. To better study the distribution of the watermarked and clean data distributions, we adopt the Wasserstein distance ([Arjovsky et al., 2017](#)) to optimize both $\mathcal{G}$ and $\mathcal{D}$. The loss functions can be written as:

$$L_{\mathcal{D}} = -\mathbb{E}_{\hat{x} \in \hat{\mathcal{X}}} \mathcal{D}(\hat{x}) + \mathbb{E}_{x \in \mathcal{X}} \mathcal{D}(\mathcal{G}(x)) + w_{\mathcal{D}} \mathbb{E}_{\hat{x} \in \hat{\mathcal{X}}, x \in \mathcal{X}} \nabla_{\alpha x + (1-\alpha)\hat{x}} \mathcal{D}(\alpha x + (1-\alpha)\hat{x}),$$

$$L_{\mathcal{G}_{\mathcal{D}}} = -w_{\mathcal{G}} \mathbb{E}_{x \in \mathcal{X}} \mathcal{D}(\mathcal{G}(x)).$$

where $w_{\mathcal{D}}$ and $w_{\mathcal{G}}$ are weights for losses and $\alpha$ is a random variable between 0 and 1[6]. On the other hand, to guarantee the quality of generated images, we adopt several loss functions to restrict the image quality, which can be written as:

$$L_x = \mathbb{E}_{x \in \mathcal{X}}[\text{L}_1(\mathcal{G}(x), x) + \text{MSE}(\mathcal{G}(x), x) + \text{LPIPS}(\mathcal{G}(x), x)].$$

where $\text{L}_1$ is the $L_1$-norm, MSE is the mean squared error loss, and LPIPS is the perceptual loss ([Zhang et al., 2018](#)).

For watermark forge, $\mathcal{G}$ is to modify $\hat{x}_i$, and $\mathcal{D}$ is to judge the distribution similarity between $\mathcal{G}(\hat{x}_i)$ and $x_i$. Therefore, the loss function can be written as:

$$L_{\mathcal{D}} = -\mathbb{E}_{x \in \mathcal{X}} \mathcal{D}(x) + \mathbb{E}_{\hat{x} \in \hat{\mathcal{X}}} \mathcal{D}(\mathcal{G}(\hat{x})) + w_{\mathcal{D}} \mathbb{E}_{\hat{x} \in \hat{\mathcal{X}}, x \in \mathcal{X}} \nabla_{\alpha x + (1-\alpha)\hat{x}} \mathcal{D}(\alpha x + (1-\alpha)\hat{x}),$$

$$L_{\mathcal{G}_{\mathcal{D}}} = -w_{\mathcal{G}} \mathbb{E}_{\hat{x} \in \hat{\mathcal{X}}} \mathcal{D}(\mathcal{G}(\hat{x})),$$

$$L_x = \mathbb{E}_{\hat{x} \in \hat{\mathcal{X}}}[\text{L}_1(\mathcal{G}(\hat{x}), \hat{x}) + \text{MSE}(\mathcal{G}(\hat{x}), \hat{x}) + \text{LPIPS}(\mathcal{G}(\hat{x}), \hat{x})].$$

---

[6]We slightly modify the discriminator loss for large-resolution images to stabilize the training process. The details can be found in Appendix A.

| Bit Length | Original | | | | | Watermark Remove | | | | | Watermark Forge | | | | |
|---|---|---|---|---|---|---|---|---|---|---|---|---|---|---|---|
| | Bit Acc | FID | PSNR | SSIM | CLIP | Bit Acc | FID | PSNR | SSIM | CLIP | Bit Acc | FID | PSNR | SSIM | CLIP |
| 4 bit | 100.00% | 4.22 | 27.81 | 0.89 | 0.99 | 52.53% | 16.36 | 24.51 | 0.86 | 0.92 | 95.76% | 17.59 | 26.70 | 0.88 | 0.94 |
| 8 bit | 100.00% | 6.19 | 25.23 | 0.83 | 0.99 | 47.80% | 18.42 | 23.59 | 0.83 | 0.91 | 97.84% | 21.09 | 24.94 | 0.82 | 0.93 |
| 16 bit | 100.00% | 11.34 | 22.71 | 0.73 | 0.98 | 50.10% | 24.63 | 23.44 | 0.77 | 0.91 | 92.23% | 18.34 | 25.84 | 0.83 | 0.94 |
| 32 bit | 99.99% | 28.76 | 19.99 | 0.53 | 0.96 | 53.64% | 25.33 | 21.17 | 0.64 | 0.91 | 90.14% | 31.13 | 23.41 | 0.71 | 0.93 |

Table 1: Performance of `WMaGi` under different bit lengths. The number of images for the adversary is 25,000.

| # of Samples (bit length = 8bit) | Original | | | | | Watermark Remove | | | | | Watermark Forge | | | | |
|---|---|---|---|---|---|---|---|---|---|---|---|---|---|---|---|
| | Bit Acc | FID | PSNR | SSIM | CLIP | Bit Acc | FID | PSNR | SSIM | CLIP | Bit Acc | FID | PSNR | SSIM | CLIP |
| 5000 | | | | | | 49.42% | 20.75 | 24.64 | 0.83 | 0.92 | 96.11% | 18.86 | 24.36 | 0.83 | 0.93 |
| 10000 | | | | | | 50.68% | 23.76 | 24.31 | 0.82 | 0.90 | 98.63% | 15.68 | 24.70 | 0.81 | 0.94 |
| 15000 | 100.00% | 6.19 | 25.23 | 0.83 | 0.99 | 59.88% | 20.32 | 22.87 | 0.80 | 0.92 | 97.80% | 25.34 | 24.55 | 0.80 | 0.92 |
| 20000 | | | | | | 54.59% | 22.90 | 24.93 | 0.84 | 0.90 | 95.99% | 23.56 | 23.74 | 0.80 | 0.92 |
| 25000 | | | | | | 47.80% | 18.42 | 23.59 | 0.83 | 0.91 | 97.84% | 21.09 | 24.94 | 0.82 | 0.93 |

Table 2: Performance of `WMaGi` under the different number of collected images. The length of embedded bits is 8.

The overall training loss for $\mathcal{G}$ can be written as

$$L_{\mathcal{G}} = L_{\mathcal{G}_D} + w_x L_x,$$

where $w_x$ is a weight for the loss function.

## 4 EVALUATIONS

### 4.1 EXPERIMENT SETUP

**Datasets.** We mainly consider two datasets: CIFAR-10 and CelebA (Liu et al., 2015). CIFAR-10 contains 50,000 training images and 10,000 test images with a resolution of 32*32. CelebA is a celebrity faces dataset, which contains 162,770 images for training and 19,867 for testing, resized at a resolution of 64*64 in our experiments. We randomly split the CIFAR-10 training set into two disjoint parts, one of which is to train the service provider's model and another is used by the adversary. Similarly, we randomly pick 100,000 images for the service provider and 10,000 images for the adversary from the training set of CelebA.

**Watermarking Schemes.** Considering the watermark's expandability to multiple users, we mainly adopt the post hoc manner, i.e., adding user-specific watermarks to the generated images. We adopt StegaStamp (Tancik et al., 2020), a state-of-the-art and robust method for embedding bit strings into given images, which is proved to be the most effective watermarking embedding method against various removal attacks (Zhao et al., 2023a). We also provide two case studies to explore the prior manner, which directly generates images with watermarks For our case studies. We follow previous works (Fei et al., 2022; Zhao et al., 2023b) to embed a secret watermark to WGAN-div (Wu et al., 2018) and EDM (Karras et al., 2022), respectively.

**Baselines.** To the best of our knowledge, `WMaGi` is the first work to remove or forge a watermark in images under a pure black-box threat model. Therefore, we consider some potential baseline attack methods under the same assumptions and attacker's capability, i.e., having only watermarked images. These baseline methods can be classified into two groups. (1) Image transformation methods: we consider modifying the properties of the given image, such as resolution, brightness, and contrast. We also consider image compression (e.g., JPEG) and image disruptions (e.g., Gaussian blurring, adding Gaussian noise). (2) Diffusion model methods (Li, 2023): we directly adopt a pre-trained unconditional diffusion model (DiffPure (Nie et al., 2022)) to modify the given image, which does not require to train a diffusion model from scratch and does not need clean images. As the diffusion model is not trained or fine-tuned for watermark removal or forge, for consistency, we do not adopt guided diffusion models or conditional diffusion models as Li (2023) did. The results from pre-trained diffusion models are various on different datasets, which will be discussed in Appendix C. Specifically, for watermark removal, the watermarked images are inputs for the attacks; for watermark forge, the clean images are inputs for the attacks.

**Implementation.** We adopt DiffPure (Nie et al., 2022) as the diffusion model used in the second step of `WMaGi` without any fine-tuning. As the adversary does not have any knowledge of the watermarking scheme, it is important to decide which checkpoint should be used in the attack. We provide a simple way to help the adversary select a checkpoint during the training process in Appendix B. More details about our implementations can be found in Appendix A, including all hyperparameters and used bit strings.

| Methods | Original | | | | | Watermark Remove | | | | | Watermark Forge | | | | |
|---|---|---|---|---|---|---|---|---|---|---|---|---|---|---|---|
| | Bit Acc | FID | PSNR | SSIM | CLIP | Bit Acc | FID | PSNR | SSIM | CLIP | Bit Acc | FID | PSNR | SSIM | CLIP |
| CenterCrop | | | | | | 59.89% | - | - | - | 0.90 | 48.33% | - | - | - | 0.93 |
| GaussianNoise | | | | | | 99.92% | 53.80 | 24.97 | 0.71 | 0.86 | 52.28% | 47.07 | 28.64 | 0.75 | 0.89 |
| GaussianBlur | | | | | | 100.00% | 25.09 | 26.26 | 0.84 | 0.86 | 52.10% | 21.18 | 28.17 | 0.88 | 0.89 |
| JPEG | | | | | | 99.27% | 17.42 | 28.40 | 0.89 | 0.89 | 52.19% | 9.96 | 33.36 | 0.94 | 0.90 |
| Brightness | | | | | | 100.00% | 4.26 | 19.70 | 0.87 | 0.95 | 52.28% | 0.39 | 21.16 | 0.91 | 0.98 |
| Gamma | 100.00% | 4.25 | 30.7 | 0.94 | 0.96 | 100.00% | 4.43 | 22.93 | 0.88 | 0.96 | 52.32% | 0.26 | 25.71 | 0.93 | 0.99 |
| Hue | | | | | | 99.99% | 5.93 | 26.84 | 0.93 | 0.94 | 52.21% | 1.60 | 32.06 | 0.98 | 0.97 |
| Contrast | | | | | | 100.00% | 4.26 | 24.28 | 0.85 | 0.95 | 52.33% | 0.25 | 27.62 | 0.90 | 0.98 |
| $DM_s$ | | | | | | 67.82% | 73.30 | 20.61 | 0.62 | 0.69 | 48.78% | 68.91 | 20.89 | 0.64 | 0.70 |
| $DM_l$ | | | | | | 47.20% | 82.38 | 15.76 | 0.34 | 0.67 | 45.96% | 79.06 | 15.81 | 0.34 | 0.68 |
| WMaGi | | | | | | 51.98% | 9.93 | 26.61 | 0.91 | 0.90 | 99.11% | 8.75 | 24.92 | 0.90 | 0.92 |

Table 3: Results of different attacks. The bit string length is 32 bits.

| Methods | Original | | | | | Watermark Remove | | | | | Watermark Forge | | | | |
|---|---|---|---|---|---|---|---|---|---|---|---|---|---|---|---|
| | Bit Acc | FID | PSNR | SSIM | CLIP | Bit Acc | FID | PSNR | SSIM | CLIP | Bit Acc | FID | PSNR | SSIM | CLIP |
| CenterCrop | | | | | | 60.53% | - | - | - | 0.87 | 50.12% | - | - | - | 0.93 |
| GaussianNoise | | | | | | 99.72% | 62.36 | 23.39 | 0.68 | 0.83 | 51.71% | 47.18 | 28.64 | 0.75 | 0.89 |
| GaussianBlur | | | | | | 100.00% | 33.23 | 24.84 | 0.81 | 0.85 | 52.06% | 21.18 | 28.17 | 0.88 | 0.89 |
| JPEG | | | | | | 99.30% | 28.94 | 25.87 | 0.85 | 0.85 | 51.74% | 9.96 | 33.36 | 0.94 | 0.90 |
| Brightness | | | | | | 100.00% | 13.37 | 18.96 | 0.83 | 0.91 | 51.83% | 0.39 | 21.16 | 0.91 | 0.98 |
| Gamma | 100.00% | 13.59 | 27.13 | 0.90 | 0.93 | 100.00% | 13.68 | 21.72 | 0.84 | 0.92 | 51.92% | 0.26 | 25.71 | 0.93 | 0.99 |
| Hue | | | | | | 99.87% | 15.82 | 24.79 | 0.88 | 0.90 | 51.86% | 1.60 | 32.06 | 0.98 | 0.97 |
| Contrast | | | | | | 100.00% | 13.66 | 22.84 | 0.82 | 0.91 | 51.85% | 0.25 | 27.62 | 0.90 | 0.98 |
| $DM_s$ | | | | | | 71.54% | 78.67 | 20.21 | 0.60 | 0.64 | 49.35% | 69.09 | 20.92 | 0.64 | 0.71 |
| $DM_l$ | | | | | | 53.75% | 82.94 | 15.67 | 0.33 | 0.67 | 50.99% | 81.66 | 15.82 | 0.34 | 0.68 |
| WMaGi | | | | | | 54.36% | 19.98 | 25.29 | 0.88 | 0.88 | 94.61% | 12.14 | 23.04 | 0.87 | 0.90 |

Table 4: Results of different attacks. The bit string length is 48 bits.

**Metrics.** To fairly evaluate our proposed WMaGi, we consider five metrics to measure its performance from different perspectives. To determine the quality of the watermark removal (forge) task, we adopt **Bit Acc**, which can be calculated as $\text{Bit Acc}(m, m') = \frac{|m| - H(m, m')}{|m|} \times 100\%$, where $H(\cdot, \cdot)$ is the Hamming Distance. If $\text{Bit Acc}(m, m') \geq \tau$, which is defined in Definition 1, verification will pass. Otherwise, it will fail. To evaluate the quality of the images generated by WMaGi and the baselines, we adopt the Fréchet Inception Distance (FID) (Heusel et al., 2017), the peak signal-to-noise ratio (PSNR) (Horé & Ziou, 2010), and the structural similarity index (SSIM) (Horé & Ziou, 2010). Furthermore, we consider the semantic information inside the images, which is evaluated by CLIP (Radford et al., 2021). For the FID, PSNR, SSIM, and CLIP scores, we compute the results between clean images and watermarked images for the watermarking scheme, and between clean images and images after removal or forge attacks.

### 4.2 ABLATION STUDY

In this section, we explore the generalizability of our proposed WMaGi under the views of the length of the embedding bits and the number of collected images. In Table 1, we show the results of WMaGi at different lengths of embedded bits. The results indicate that WMaGi is robust for different secret message lengths. Specifically, when the length of the embedded bits increases, WMaGi can still achieve good performance on watermark removing or forging and make the transferred images keep high quality and maintain semantic information. In Table 2, we present the results when the adversary uses the different numbers of collected images as his training data. The results indicate that even with limited data, the adversary can remove or forge a specific watermark without harming the image quality, which proves that our method can be a real-world threat. Therefore, our proposed WMaGi has outstanding flexibility and generalizability under a practical threat model. We further prove its extraordinary few-shot generalizability for unseen watermarks in Section 4.3.

### 4.3 MAIN RESULTS ON POST HOC MANNERS

In this section, we focus on post hoc manners, i.e., adding watermarks to AIGC with an embedding model. Because the post hoc watermarking scheme can freely change the embedding watermarks, we evaluate WMaGi under few-shot learning to show the capability of adapting to unseen watermarks.

**Results on CelebA.** We consider two different lengths of the embedding bits, i.e., 32-bit and 48-bit. Furthermore, we do not consider the specific coding scheme, including the source coding and the channel coding. In Tables 3 and 4, we compare the results of WMaGi and the baseline methods on the watermark removal task and the watermark forging task, respectively. We notice that the watermark embedding method is robust against various image transformations. Using image transformations cannot simply remove or forge a specific watermark in the given images. For methods using diffusion models, we consider two settings, i.e., adding large noise to the input ($DM_l$) and adding small noise

| # of Samples | Original | | | | | Watermark Remove | | | | | Watermark Forge | | | | |
|---|---|---|---|---|---|---|---|---|---|---|---|---|---|---|---|
| (bit length = 32bit) | Bit Acc | FID | PSNR | SSIM | CLIP | Bit Acc | FID | PSNR | SSIM | CLIP | Bit Acc | FID | PSNR | SSIM | CLIP |
| 10 | | | | | | 49.98% | 46.90 | 23.19 | 0.81 | 0.83 | 72.64% | 12.27 | 22.43 | 0.89 | 0.91 |
| 50 | 100.00% | 4.14 | 30.69 | 0.94 | 0.96 | 53.31% | 19.74 | 24.47 | 0.87 | 0.86 | 83.18% | 11.89 | 28.37 | 0.94 | 0.93 |
| 100 | | | | | | 53.27% | 14.30 | 25.51 | 0.89 | 0.87 | 93.47% | 12.43 | 26.57 | 0.92 | 0.91 |

Table 5: Few-shot generalization ability of `WMaGi` on unseen watermarks.

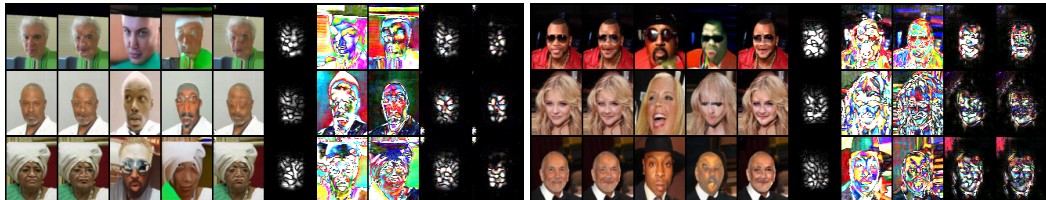

(a) Watermark Removal  (b) Watermark Forging

Figure 2: The first column is clean images. The second is watermarked images. The third is the output of $\mathrm{DM}_l$. The fourth is the output of $\mathrm{DM}_s$. The fifth is the output of `WMaGi`. The sixth is the difference between the first and second columns. The seventh is the difference between the first and third columns. The eighth is the difference between the first and fourth columns. The ninth is the difference between the first and fifth columns. The tenth is the difference between the second and fifth columns.

to the input ($\mathrm{DM}_s$). Especially, we use the same setting as $\mathrm{DM}_l$ in the second step of `WMaGi` to generate images. Although diffusion models can easily remove the watermark from the given images under both settings, the generated images are visually different from the input images, causing a low PSNR, SSIM, and CLIP score. Furthermore, the FID indicates that the diffusion model will cause a distribution shift compared to the clean dataset. Nevertheless, we find that $\mathrm{DM}_l$ and $\mathrm{DM}_s$ can maintain high image quality while successfully removing watermarks on other datasets, which we discuss in Appendix C. The results make us reflect on the generalizability of diffusion models on different datasets and watermarking schemes. However, evaluating all accessible diffusion models on various datasets and watermarking schemes will take months. Therefore, we leave it as future work to deeply study the diffusion models in the watermarking removal task. On the other hand, forging a specific unknown watermark is non-trivial and impossible for both image transformation methods and diffusion models.

Our `WMaGi` gives an outstanding performance in both tasks and maintains good image quality as well. However, we notice that as the length of the embedded bit string increases, it becomes more challenging to forge or remove the watermark. That is the reason that under 48-bit length, our `WMaGi` has a little performance drop on both tasks with respect to bit accuracy and image quality. We provide visualization results in the following content to prove images generated by `WMaGi` are still visually close to the given image under a longer embedding length. More importantly, `WMaGi` is very time-efficient compared to diffusion model methods. We present the results in Appendix D.

**Few-Shot Generalization.** In real-world applications, large companies can assign a unique watermark for every account or change watermarks periodically. Therefore, it is important to study the few-shot power of `WMaGi`, i.e., fine-tuning `WMaGi` with several new data with an unseen watermark to achieve outstanding watermark removal or forging abilities for the unseen watermark. In our experiments, we mainly consider embedding a 32-bit string into clean images. Then, we fine-tune the model in Table 3 to fit new unseen watermarks. In Table 5, we present the results under 10, 50, and 100 training data for watermark removal and forging. The results indicate that the watermark removal task is much easier than the watermark forging task. Furthermore, with more accessible data, both bit accuracy and image quality can be improved. It is worth noticing that, even with limited data, `WMaGi` can successfully remove or forge an unseen watermark and maintain high image quality. The results prove that our proposed method has strong few-shot generalization power to meet practical usage.

**Visualization.** To better compare the image quality of `WMaGi` with other baselines, we show the visualization results in Figure 2. Specifically, both $\mathrm{DM}_s$ and $\mathrm{DM}_l$ will change the semantic information in inputs. `WMaGi` can keep the image details in the watermark removal and forging tasks. Furthermore, when comparing the differences between clean and watermarked images, we find that `WMaGi` can produce a similar residual as the watermark embedding model, which means that `WMaGi` can learn the embedding information during the training process. More results can be found in Appendix E.

| Methods | WGAN-div | | | | | | EDM | | | | | |
|---|---|---|---|---|---|---|---|---|---|---|---|---|
| | Original | | Watermark Remove | | Watermark Forge | | Original | | Watermark Remove | | Watermark Forge | |
| | Bit Acc | FID | Bit Acc | FID | Bit Acc | FID | Bit Acc | FID | Bit Acc | FID | Bit Acc | FID |
| CenterCrop | | | 58.05% | - | 48.25% | - | | | 61.31% | - | 50.23% | - |
| GaussianNoise | | | 99.09% | 100.65 | 52.24% | 41.58 | | | 82.20% | 56.43 | 50.50% | 50.98 |
| GaussianBlur | | | 99.59% | 56.83 | 52.04% | 22.85 | | | 64.10% | 40.04 | 51.42% | 36.49 |
| JPEG | | | 98.43% | 64.09 | 52.30% | 15.12 | | | 52.90% | 26.81 | 49.89% | 21.21 |
| Brightness | | | 99.65% | 56.99 | 52.14% | 0.63 | | | 99.43% | 8.30 | 51.15% | 0.65 |
| Gamma | 99.66% | 60.20 | 99.66% | 60.59 | 52.25% | 0.47 | 99.99% | 8.68 | 99.93% | 9.12 | 51.16% | 0.37 |
| Hue | | | 99.55% | 63.70 | 52.17% | 1.83 | | | 99.86% | 8.96 | 50.92% | 2.16 |
| Contrast | | | 99.66% | 57.47 | 52.27% | 0.32 | | | 99.79% | 50.98 | 51.11% | 0.39 |
| $DM_s$ | | | 67.12% | 100.93 | 49.17% | 68.79 | | | 51.03% | 78.08 | 51.14% | 79.75 |
| $DM_l$ | | | 47.16% | 117.80 | 46.20% | 83.36 | | | 51.69% | 58.39 | 51.31% | 60.00 |
| WMaGi | | | 52.12% | 69.88 | 95.72% | 5.84 | | | 64.56% | 19.58 | 90.75% | 5.98 |

Table 6: Results of removing and forging content watermarks from the WGAN-div and EDM.

## 4.4 MAIN RESULTS ON PRIOR MANNERS

In this section, we focus on prior methods, i.e., directly embedding watermarks into the generative models. We follow the previous methods (Fei et al., 2022) and (Zhao et al., 2023b) to embed a secret bit string into a WGAN-div and an EDM as a watermark, respectively. Therefore, all generated images contain a pre-defined watermark, but we cannot have the corresponding clean images. That is to say, we cannot obtain the PSNR, SSIM, and CLIP scores as previously. So, we only evaluate the FID and the bit accuracy in our experiments. Specifically, we train the WGAN-div with 100,000 watermarked images randomly selected from the training set of CelebA. We directly use the models provided by Zhao et al. (2023b), which are trained on FFHQ embedded with a 64-bit string. For WMaGi, we use the WGAN-div and EDM to generate 10,000 samples as the accessible data.

In Table 6, we show the results of different attacks to remove or forge the watermark. First, we find that embedding a watermark into the generative model will cause the generated images to have a different distribution from the clean images, making the FID extremely high. Second, the EDM can generate high-quality images even under watermarking, causing a lower FID. However, we find that the embedded watermark by Zhao et al. (2023b) is less robust, which can be removed by blurring and JPEG compression. It could be because they made some trade-off between the image quality and robustness. For both, WMaGi can successfully remove and forge the specific watermark in the generated images and keep the same image quality as the generative model. The visualization results can be found in Appendix E.

## 4.5 POTENTIAL DEFENSES FOR SERVICE PROVIDERS

Although WMaGi is an effective method for removing or forging a specific watermark in images, there are some possible defense methods against our attack. First, large companies can assign a group of watermarks to an account to identify the identity. When adding watermarks to the images, the watermark can be randomly selected from the group of watermarks, which can hinder the adversary from obtaining images containing the same watermark. However, such a method requires a longer length of embedded watermarks to meet the population of users, which will decrease the image quality because embedding a longer watermark will damage the image. Another defense is to design a more robust watermarking scheme, which can defend against removal attacks from diffusion models. Because WMaGi requires diffusion models to remove the watermarks. The aforementioned two methods have the potential to defend against WMaGi but have different shortcomings, such as decreasing the image quality, requiring a newly designed coding scheme, and requiring a newly designed robust watermarking scheme. Therefore, WMaGi will be a threat for future years.

## 5 LIMITATIONS AND CONCLUSIONS

In this paper, we consider a practical threat to AIGC protection and regulation schemes, which are based on the state-of-the-art watermarking technology. We introduce WMaGi, a unified attack framework to effectively remove or forge watermarks over AIGC while maintaining good image quality. With WMaGi, the adversary only requires watermarked images without their corresponding clean ones, making it a real-world threat. Through comprehensive experiments, we prove that WMaGi has strong few-shot generalization abilities to fit unseen watermarks, which makes it more powerful.

In Appendix F, we discuss the limitations of WMaGi for larger-resolution and more complex images. Although WMaGi can still successfully forge the watermark to some degree, the performance is not good enough for a real-world scenario. However, we believe that further improvement over WMaGi is probable with more advanced GAN structures and training strategies. In Appendix G, we discuss the social impact of our work. WMaGi brings both positive and negative impacts on the society.

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

# A    EXPERIMENT SETTINGS

**Model Structures.** For CIFAR-10 and CelebA, we choose different architectures for generators and discriminators to stabilize the training process. Specifically, when training models on CIFAR-10, we use the ResNet-based generator architecture (Zhu et al., 2017) with 6 blocks. As the CelebA images have higher resolution, we use the ResNet-based generator architecture (Zhu et al., 2017) with 9 blocks. For the discriminators, we use a simple model containing 4 convolutional layers for CIFAR-10. And for CelebA, a simple discriminator cannot promise a stable training process. Therefore, we use a ResNet-18 (He et al., 2016). To improve the quality of generated images, we follow the residual training manner, that is, the output from the generators will be added to the original input.

**Hyperparameters.** We use different hyperparameters for CIFAR-10 and CelebA, respectively. When training models on CIFAR-10, we use RMSprop as the optimizer for both the generator and the discriminator. The learning rate is 0.0001, and the batch size is 32. We set $w_{\mathcal{D}} = 10$, and the total number of training epochs is 1,000. We update the generator's parameters after 5 times of updating of the discriminator's parameters. For CelebA, we adopt Adam as our model optimizer. The learning rate is 0.003, and the batch size is 16. We replace the discriminator loss with the one from StyleGAN (Karras et al., 2019) with $w_{\mathcal{D}} = 5$, and the total number of training epochs is 1,000. We update the generator's parameters after updating the discriminator's parameters. We present $w_{\mathcal{G}}$ and $w_x$ in Table 7 used in our experiments. We choose the best model based on the image quality.

**Baseline Settings.** For image transformation methods, we mainly adopt `torchvision` to implement attacks. To adjust brightness, contrast, and gamma, the changing range is randomly selected from 0.5 to 1.5. To adjust the hue, the range is randomly selected from -0.1 to 0.1. For center-cropping, we randomly select the resolution from 32 to 64. For the Gaussian blurring, we randomly choose the Gaussian kernel size from 3, 5, and 7. For adding Gaussian noise, we randomly choose $\sigma$ from 0.0 to 0.1. For JPEG compression, we randomly selected the compression ratio from 50 to 100. When evaluating the results of image transformation methods, we run multiple times and use the average results. For diffusion methods $DM_l$, we set the sample step as 30 and the noise scale as 150. For diffusion methods $DM_s$, we set the sample step as 200 and the noise scale as 10. Specifically, we use $DM_l$ in the second step of `WMaGi`. Considering using diffusion models to generate images is very time-consuming, we randomly select 1,000 images from the test set to obtain the results for diffusion models.

**Embedded Bits.** In Table 8, we list the bit strings embedded in the images in our experiments.

| Experiment | Watermark Remove | | Watermark Forge | |
|---|---|---|---|---|
|  | $w_{\mathcal{G}}$ | $w_x$ | $w_{\mathcal{G}}$ | $w_x$ |
| CIFAR-10 4bit | 500 | 10 | 500 | 5 |
| CIFAR-10 8bit | 800 | 15 | 500 | 10 |
| CIFAR-10 16bit | 500 | 40 | 150 | 40 |
| CIFAR-10 32bit | 100 | 40 | 100 | 40 |
| CIFAR-10 5000 data | 800 | 15 | 500 | 10 |
| CIFAR-10 10000 data | 800 | 15 | 600 | 20 |
| CIFAR-10 15000 data | 500 | 15 | 500 | 10 |
| CIFAR-10 20000 data | 800 | 15 | 500 | 15 |
| CIFAR-10 25000 data | 800 | 15 | 500 | 10 |
| CelebA 32bit | 10 | 120 | 1 | 10 |
| CelebA 48bit | 10 | 200 | 1 | 10 |
| Few-Shot 10 Images | 10 | 200 | 1 | 10 |
| Few-Shot 50 Images | 10 | 200 | 1 | 10 |
| Few-Shot 100 Images | 10 | 200 | 1 | 10 |
| WGAN-div | 10 | 120 | 1 | 10 |
| EDM | 1 | 10 | 100 | 1 |

Table 7: Hyperparameter settings in our experiments for watermark removal and watermark forging.

| Experiment | Bit String |
|---|---|
| CIFAR-10 4bit | 1000 |
| CIFAR-10 8bit | 10001000 |
| CIFAR-10 16bit | 1000100010001000 |
| CIFAR-10 32bit | 10001000100010001000100010001000 |
| CelebA 32bit | 10001000100010001000100010001000 |
| CelebA 48bit | 100010001000100010001000100010001000100010001000 |
| Few-Shot | 111000111010101010000100000001011 |
| WGAN-div | 10001000100010001000100010001000 |
| EDM | 010001000100001011101011111111001110100000111110110101010110000000 |

Table 8: Selected bit strings in our experiments.

## B SELECT A CORRECT CHECKPOINT

It is important to choose the correct checkpoint because it is closely associated with the attack performance. However, when the adversary does not have any information about the watermarking scheme, it is unavailable to determine the best checkpoint with Bit Acc as metrics. However, after plotting the bit accuracy in Figure 3, we find that the performances of different checkpoints in the later period are close and acceptable for a successful attack under the Bit ACC metrics. Therefore, we choose the best checkpoint from the later training period based on the image quality metrics, including the FID, SSIM, and PSNR, in our experiments. It is to say, our selection strategy does not violate the threat model, where the adversary can only obtain watermarked images.

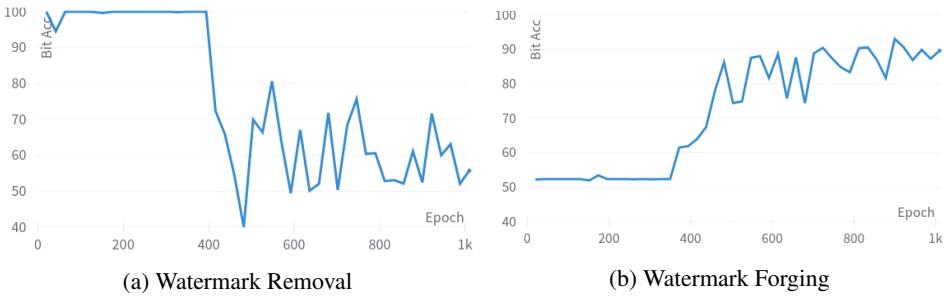

(a) Watermark Removal          (b) Watermark Forging

Figure 3: Bit Acc for different tasks during the training stage on CelebA.

## C DIFFUSION MODELS FOR WATERMARK REMOVAL

In our experiments, we find that the pre-trained diffusion models will not promise a similar output as the input image without the guidance on CelebA. However, when we evaluate the diffusion models on another dataset, LSUN-bedroom (Yu et al., 2015), we find that even under a very large noise scale, the output of the diffusion model is very close to the input image, and the watermark has been successfully removed. The visualization results can be found in Figure 4, where we use 30 sample steps and 150 noise scales for $DM_l$ and use 200 sample steps and 10 noise scales for $DM_s$, which are the same as the settings on CelebA. The numerical results in Table 9 prove that the diffusion model can maintain high image quality under large inserted noise.

| Diffusion Model Setting (bit length = 32bit) | | Original | | | | | Watermark Remove | | | | |
|---|---|---|---|---|---|---|---|---|---|---|---|
| Sample Step | Noise Scale | Bit Acc | FID | PSNR | SSIM | CLIP | Bit Acc | FID | PSNR | SSIM | CLIP |
| 30 | 150 | | | | | | 51.81% | 75.52 | 20.15 | 0.58 | 0.88 |
| 50 | 150 | | | | | | 51.50% | 84.14 | 18.92 | 0.55 | 0.86 |
| 100 | 150 | 100.00% | 10.67 | 39.49 | 0.98 | 0.99 | 50.47% | 95.27 | 16.69 | 0.49 | 0.83 |
| 200 | 10 | | | | | | 56.16% | 73.01 | 22.11 | 0.72 | 0.84 |
| 200 | 30 | | | | | | 53.03% | 98.00 | 19.37 | 0.59 | 0.80 |
| 200 | 50 | | | | | | 53.81% | 108.71 | 17.63 | 0.52 | 0.78 |

Table 9: Numerical results of watermark removal with diffusion models under different noise scales and sample steps.

We think the performance differences on CelebA and LSUN are related to the resolution and image distribution. Specifically, images in CelebA are 64 * 64 and only contain human faces. The diversity

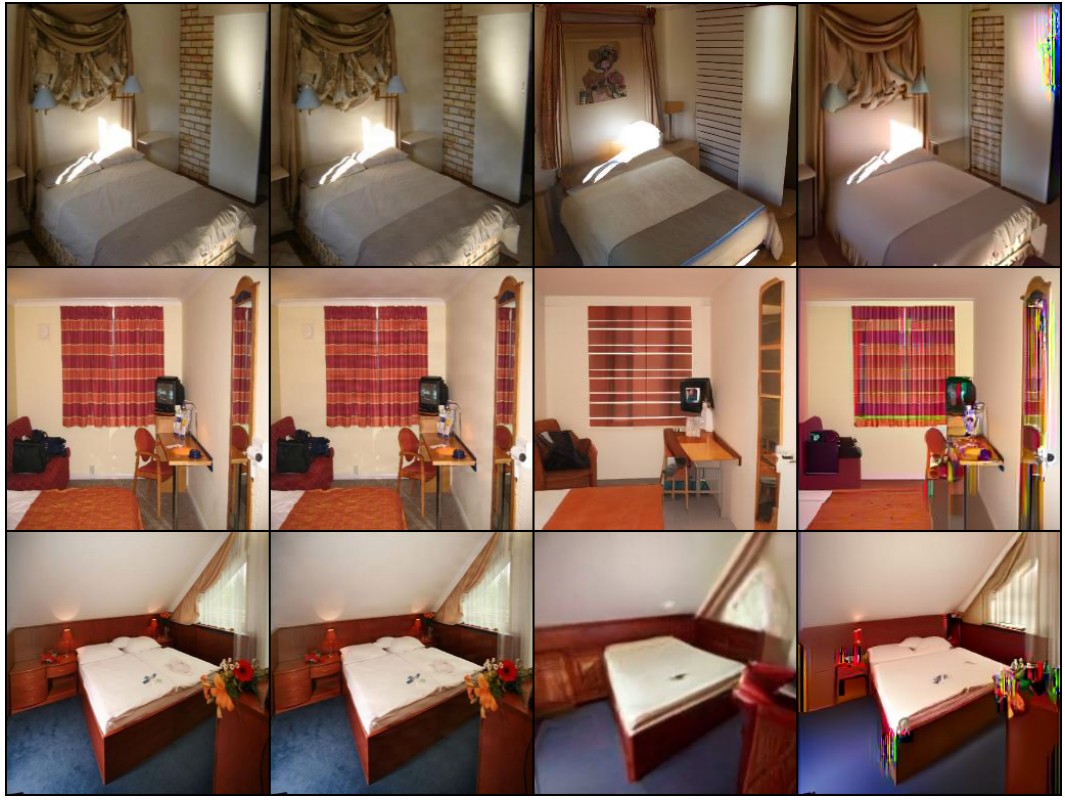

Figure 4: The first column is clean images. The second is watermarked images. The third is the output of $DM_l$. The fourth is the output of $DM_s$.

of faces is not too high. However, images in LSUN are 256 * 256 and have different decoration styles, illumination, and perspective, which means the diversity of bedrooms is very high. Therefore, transforming an image into another image in LSUN is more challenging than doing that in CelebA. This could be the reason that diffusion models cannot produce an output similar to that of CelebA. This limitation is critical for an attack based on diffusion models. Therefore, we appeal to comprehensively evaluate the performance of the watermark removal task for various datasets.

## D    TIME COST VS DIFFUSION MODELS

To compare the time cost for generating one image with a given one, we record the total time cost for 1,000 images on one A100. The batch size is fixed to 128. For $DM_l$, the total time cost is 5,231.72 seconds. For $DM_s$, the total time cost is 2325.01 seconds. For `WMaGi`, the total time cost is **0.46** seconds. Therefore, our method is very fast and efficient.

## E    OTHER VISUALIZATION RESULTS

In this section, we show the other visualization results in our experiments. In Figure 5, we present the visualization results for the few-shot experiments. The results indicate that with more training samples, image quality can be improved. And, even with a few samples, `WMaGi` can learn the embedding pattern.

In Figures 6 and 7, we show the visualization results of WGAN-div and EDM, respectively. As the generated images from WGAN-div are affected by the watermarking, the images after removal contain some noise. However, the watermark forging can keep high image quality on the clean images. Furthermore, EDM can generate higher-quality images. Therefore, the watermarked images after removal will obtain less noise.

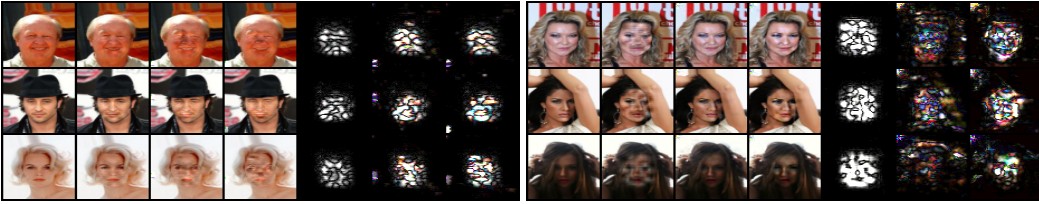

(a) Watermark Removal           (b) Watermark Forging

Figure 5: The first column is clean images. The second is watermarked images. The third is the output of `WMaGi` under the 50-sample setting in the few-shot experiment. The fourth is the output of `WMaGi` under the 100-sample setting in the few-shot experiment. The fifth is the difference between the first and second columns. The sixth is the difference between the first and third columns. The seventh is the difference between the first and fourth columns.

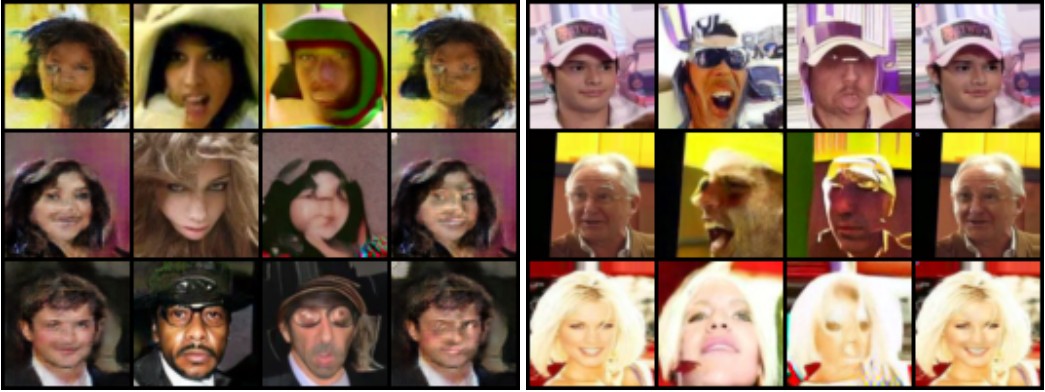

(a) Watermark Removal           (b) Watermark Forging

Figure 6: The first column is WGAN-div generated watermarked images for Figure 6a and clean images for Figure 6b. The second is the output of $DM_l$. The third is the output of $DM_s$. The fourth is the output of `WMaGi`.

## F  LIMITATIONS ON LARGE-RESOLUTION AND COMPLEX IMAGES

We focus on CelebA in our main paper, which contains human faces in a resolution of 64 * 64. In this part, we discuss the limitation of our method on larger resolution and more complex images. To evaluate our method on such images, LSUN-bedroom (Yu et al., 2015) is a good choice, in which the image resolution is 256 * 256. Similarly to the CelebA experiment settings, we randomly select 10,000 images for `WMaGi`, and the bit length is 32. As watermark removal is easy to do with only diffusion models, forging is more challenging and critical.

In Figure 8, we illustrate the bit accuracy during the training stage of `WMaGi`. Although accuracy increases with increasing training steps, we find that it is difficult to achieve accuracy over 80%. If we increase the number of training steps, the accuracy will be stable around 75%. In Figure 9, we compare the images before and after `WMaGi`. It is impossible for human eyes to figure out what are clean images, which shows that `WMaGi` can maintain impressive image quality even for large-resolution and complex images. While `WMaGi` is still effective for large-resolution and complex images, we think its ability is constrained. Our future work will be to improve its effectiveness for more complex data.

## G  SOCIAL IMPACT

We think that the proposed `WMaGi` will cause some malicious users to freely make the AIGC for commercial use. Furthermore, malicious users could frame their users by spreading illegal AIGC with forged watermarks. We think they are some negative impacts from our research. But, on the other hand, our work will encourage others to explore a more robust and reliable watermark for AIGC, which is a positive impact on society.

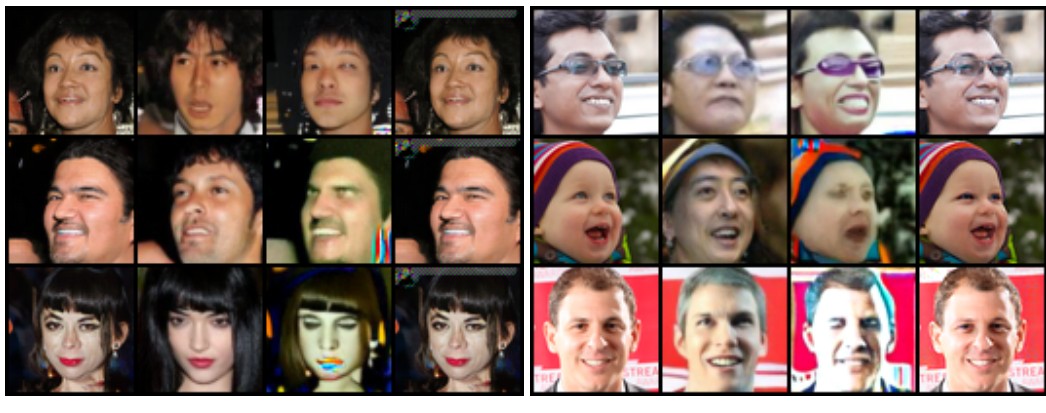

    (a) Watermark Removal                          (b) Watermark Forging

Figure 7: The first column is EDM generated watermarked images for Figure 6a and clean images for Figure 6b. The second is the output of $DM_l$. The third is the output of $DM_s$. The fourth is the output of `WMaGi`.

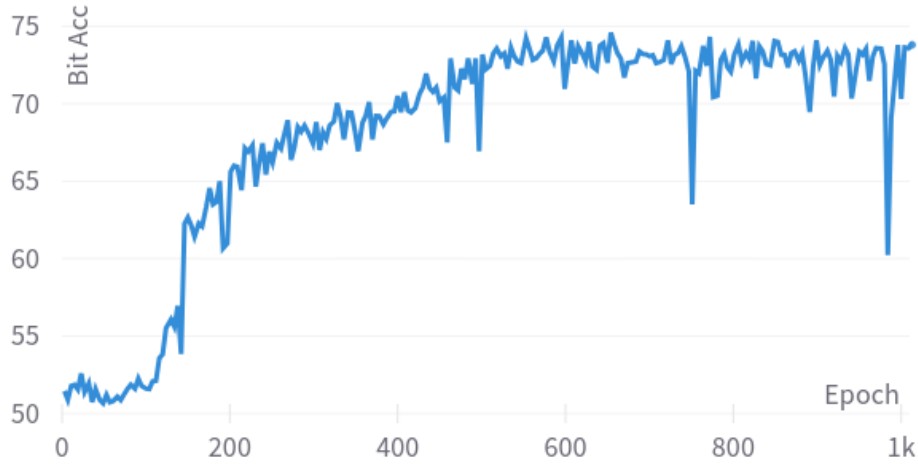

Figure 8: Bit Acc with training epoch increasing.

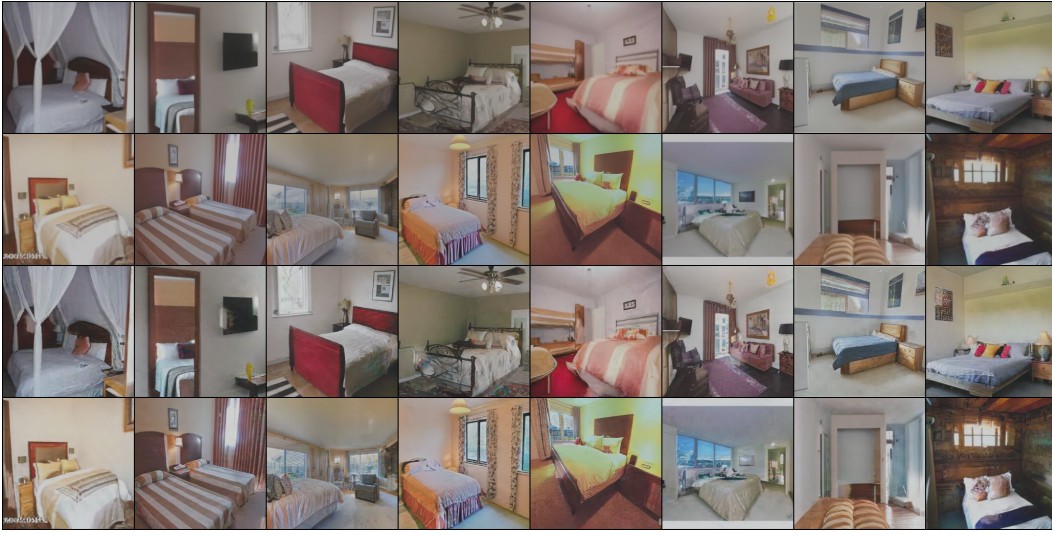

Figure 9: Clean images and corresponding outputs from `WMaGi`. The top two rows are clean images.

