# OpenReview forum: "Towards the Vulnerability of Watermarking Artificial Intelligence Generated Content"
_ICLR.cc/2024/Conference — ICLR 2024 Conference Withdrawn Submission_

### Official Review · Reviewer_Z66i · 2023-10-22

**Soundness:** 3 good
**Presentation:** 1 poor
**Contribution:** 2 fair
**Rating:** 3
**Confidence:** 4

**Summary:**

This paper introduces WMaGi, a unified attack framework to remove or forge watermarks over AIGC. However, I think the resulting image quality is not good.

**Strengths:**

- To the best of my knowledge, the proposed WMaGi is the first work to forge a watermark in images under a pure black-box threat model.

**Weaknesses:**

- I think the proposed method is impractical. I think the biggest problem lies with the fact that the proposed method damages the image when removing the watermark.
- I don't think two separate tables (i.e., Tab.3 and Tab.4) for 32 (bit string length) and 48 is a reasonable layout. If the authors focus on 32, they could put the results of 48 in the supplementary material, thus flowing more space to present other experimental phenomena.
- The results of the experiment are not clearly presented. The authors need to be aware that the reader of the paper is not necessarily familiar with the field. For example,  for PSNR, a higher value is preferable. The authors need to point this out to make it easier for the reader to understand the experimental results. Besides, the author should bold the best results in each table.

**Questions:**

Math formulas are poorly presented. In page 5, L_x, L_D, and L_gD are confused: there are two different interpretations of each loss. As a result, I cannot understand how to optimize the overall training loss G. Besides, what is the use of L_D, I wonder what role L_D plays in the final loss function?

---

### Official Review · Reviewer_K8Tz · 2023-10-31

**Soundness:** 3 good
**Presentation:** 3 good
**Contribution:** 3 good
**Rating:** 3
**Confidence:** 4

**Summary:**

The paper discusses the vulnerabilities of watermarking methods used in Artificial Intelligence Generated Content (AIGC). The authors introduce WMaGi, a unified framework that can execute watermark removal and forging attacks against AIGC. They argue that existing watermarking schemes are vulnerable, as adversaries can either erase or forge watermarks in AIGC, making the content untraceable to the original user or falsely attributed to another user. WMaGi leverages pre-trained diffusion models and generative adversarial networks (GANs) for these attacks, maintaining the quality of the generated content. The evaluation of WMaGi shows its effectiveness and efficiency in various settings, proving that it can successfully manipulate watermarks in a practical, black-box scenario, highlighting the need for more robust watermarking methods in AIGC

**Strengths:**

The third paper exhibits several strengths:

1. Comprehensive Framework (WMaGi): The paper introduces WMaGi, a unified framework that is capable of executing both watermark removal and forging attacks against AI-Generated Content (AIGC). This comprehensive approach allows for a broader evaluation of watermarking vulnerabilities.

2. Use of Pre-trained Models: WMaGi leverages pre-trained diffusion models and generative adversarial networks (GANs) for the attacks, which enhances the practical relevance and applicability of the proposed methods, ensuring that the attacks are sophisticated and aligned with current technological advancements.

3. Practical Evaluation: The paper conducts a practical evaluation of the proposed method in a black-box scenario, demonstrating its effectiveness and efficiency across various settings. This practical focus ensures that the findings are relevant and applicable to real-world watermarking challenges in AI-generated content.

**Weaknesses:**

1. Dependency on Watermark Length: The effectiveness of the WMaGi method seems to be influenced by the length of the embedded watermark. As the length of the embedded bit string increases, it becomes more challenging to forge or remove the watermark, causing a slight performance drop in terms of bit accuracy and image quality.

2. Effectiveness in Real-World Applications: While WMaGi is shown to be effective in a controlled experimental setting, its practical effectiveness in real-world applications, where watermarks might be more complex and diverse, is not thoroughly evaluated.

3. Defense Mechanisms: The paper discusses potential defense mechanisms against the WMaGi attack, such as assigning a group of watermarks to an account. These defenses could hinder the adversary from obtaining images containing the same watermark, suggesting that there might be ways to mitigate the effectiveness of the WMaGi attack

**Questions:**

1. How does the WMaGi framework perform across various watermarking schemes, such as [1] and is it universally effective against different types of embedded watermarks in AI-Generated Content (AIGC)?
2. How does the WMaGi framework perform on higher-resolution images? In general, the watermark is easier to hide in higher resolution imagessince there is usually more redundancy in those images.

[1] Pierre Fernandez, Alexandre Sablayrolles, Teddy Furon, Hervé Jégou, and Matthijs Douze. 2022. Watermarking Images in Self-Supervised Latent Spaces. In IEEE International Conference on Acoustics, Speech and Signal Processing (ICASSP).
IEEE, Singapore, 3054–3058

---

### Official Review · Reviewer_o1H9 · 2023-11-01

**Soundness:** 2 fair
**Presentation:** 3 good
**Contribution:** 2 fair
**Rating:** 5
**Confidence:** 3

**Summary:**

In this paper, the authors introduce a novel image watermark removal and forgery method named WMaGi. As a proposed black-box pipeline for executing both tasks, WMaGi first denoises the watermarked images to obtain clean and watermarked pairs. Subsequently, WMaGi employs a GAN to either remove or add watermarks. The resulting model is more effective and efficient than baseline methods.

**Strengths:**

- The paper is well-written, and the problem discussed is of significant importance.
- The threat model is clearly defined. Being a total black-box and not requiring access to clean images are notable strengths.
- The thorough ablation studies in the paper are highly appreciated.

**Weaknesses:**

- In my opinion, the quality of the output images from WMaGi is not really good. They appear very blurry. Additionally, for the EDM model shown in Figure 7, the generated outputs exhibit some artifacts.
- I believe this paper misses an important watermark mentioned in [1]. As demonstrated in [2], where a diffusion model is employed to denoise the watermark, the watermark from [1] is challenging to remove. It would be insightful to determine if WMaGi is effective against it.

[1] Wen, Y., Kirchenbauer, J., Geiping, J., & Goldstein, T. (2023). Tree-Ring Watermarks: Fingerprints for Diffusion Images that are Invisible and Robust. arXiv preprint arXiv:2305.20030.

[2] Zhao, X., Zhang, K., Wang, Y. X., & Li, L. (2023). Generative Autoencoders as Watermark Attackers: Analyses of Vulnerabilities and Threats. arXiv preprint arXiv:2306.01953.

**Questions:**

- Which DiffPure model was utilized in the experiment? Was it trained on Cifar-10 or CelebA? If it was, this could pose a problem because the attacker would have some knowledge of the target diffusion model. Would the method maintain its efficacy if the ImageNet DiffPure were employed?
- Could the attack be enhanced by employing better diffusion models for the data processing step, such as Stable Diffusion?

---

### Official Review · Reviewer_m88T · 2023-11-02

**Soundness:** 3 good
**Presentation:** 3 good
**Contribution:** 3 good
**Rating:** 5
**Confidence:** 4

**Summary:**

In this paper, the authors propose a watermark removal and forge method in the black-box setting. In detail, the proposed method first collect the data from the target service provider's API, and then adopt the diffusion model to weaken the watermark embedded. The final step is to train a GAN to remove or forge the watermark. The authors conduct the experiment on two image datasets to evaluate the effectiveness of the proposed method.

**Strengths:**

1. The developed method is under the black-box setting, shows the vulnerability of the existing watermarking method.

2. Compared with existing diffusion model based attacks, the proposed method only adopts the diffusion model to generate the data for GAN training, which improves the efficiency.

3. The proposed method has the ability of few-shot generalization to unseen watermarks.

**Weaknesses:**

1. In the methodology part, some parts are not fully discussed.

     1.1 Lack the discussion about the necessary of adopting the diffusion model to weaken the watermark embedded.

     1.2 As mentioned in step 2 of section 3.3, the proposed method does not implicitly constrain the similarity. Why not constrain the similarity? How about if it is constrained with similarity? And I guess not constraining the similarity can bring the benefits that the model adopted in step 2 can be directly applied without fine-tuning to satisfy the similarity constraint. Is it correct?

2. In the experiment part, more experiments are required.

  2.1 In section 4.5, the authors provide some potential defenses for service providers, including the group of watermarks. The authors should give the experimental evidence to support their defense suggestions.

Minor (it will not affect my review opinion, just a friendly reminder)

1. Typo: in section 4.1, the second paragraph: "For our case studies" -> "for our case studies"

2. The format of the table needs further improvement.

**Questions:**

See above weakness.